# ITERATIVELY REFINED BEHAVIOR REGULARIZATION FOR OFFLINE REINFORCEMENT LEARNING

## ABSTRACT

One of the fundamental challenges for offline reinforcement learning (RL) is ensuring robustness to data distribution. Whether the data originates from a near-optimal policy or not, we anticipate that an algorithm should demonstrate its ability to learn an effective control policy that seamlessly aligns with the inherent distribution of offline data. Unfortunately, *behavior regularization*, a simple yet effective offline RL algorithm, tends to struggle in this regard. In this paper, we propose a new algorithm that substantially enhances behavior-regularization based on *conservative policy iteration*. Our key observation is that by iteratively refining the reference policy used for behavior regularization, conservative policy update guarantees gradually improvement, while also implicitly avoiding querying out-of-sample actions to prevent catastrophic learning failures. We prove that in the tabular setting this algorithm is capable of learning the optimal policy covered by the offline dataset, commonly referred to as the *in-sample optimal* policy. We then explore several implementation details of the algorithm when function approximations are applied. The resulting algorithm is easy to implement, requiring only a few lines of code modification to existing methods. Experimental results on the D4RL benchmark indicate that our method outperforms previous state-of-the-art baselines in most tasks, clearly demonstrate its superiority over behavior regularization.

## 1 INTRODUCTION

Reinforcement learning (RL) has achieved considerable success in various decision-making problems, including games (Mnih et al., 2015), recommendation and advertising (Hao et al., 2020), logistics optimization (Ma et al., 2021) and robotics (Mandlekar et al., 2020). A critical obstacle that hinders a broader application of RL is its trial-and-error learning paradigm. For applications such as education, autonomous driving and healthcare, active data collection could be either impractical or dangerous (Levine et al., 2020). Instead of learning actively, a more favorable approach is to employ scalable data-driven learning methods that can utilize existing data and progressively improve as more training data becomes available. This motivates *offline RL*, with its primary objectivge being learning a control policy solely from previously collected data.

Offline datasets frequently offer limited coverage of the state-action space. Directly utilizing standard RL algorithms to such datasets can result in extrapolation errors when bootstrapping from out-of-distribution (OOD) state-actions, consequently causing significant overestimations in value functions. To address this, previous works impose various types of constraints to promote pessimism towards accessing OOD state-actions. One simple yet effective approach is *behavior regularization*, which penalizes significant deviations from the behavior policy that collects the offline dataset (Kumar et al., 2019; Wu et al., 2019; Siegel et al., 2020; Brandfonbrener et al., 2021; Fujimoto & Gu, 2021). For example, TD3+BC optimizes a deterministic policy $\pi$ by $\max_\pi \mathbb{E}_{(s,a)\sim\mathcal{D}}[v^\pi(s) - \lambda(\pi(s) - a)]$, where $\mathcal{D}$ is an offline dataset, $\lambda > 0$ is a hyper-parameter (Fujimoto & Gu, 2021). This encourages the learned policy to remain conservative and enhances the stability of policy update, preventing the learned policy from taking excessively risky or off-policy actions.

A fundamental challenge in behavior regularization lies in its performance being closely tied to the underlying data distribution. Previous works suggest that it often yields subpar results when applied to datasets originating from suboptimal policies (Kostrikov et al., 2022; Xiao et al., 2023). To better understand this phenomenon, we investigate how the quality of a behavior policy influences the

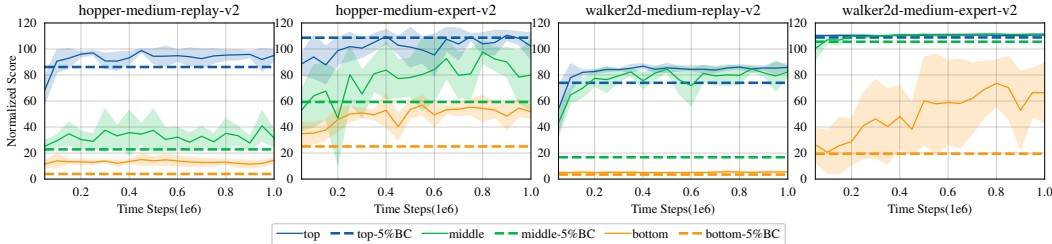

Figure 1: The dashed lines indicate the performance of reference policies trained on filtered sub-datasets consisting of top 5%, median 5%, and bottom 5% trajectories. The solid lines indicate the performance of TD3 with different reference policies. It can be concluded that behavior regularization has advantage over behavior cloning and the efficacy of behavior regularization is strongly contingent on the reference policy.

performance of behavior regularization through utilizing *percentile behavior cloning* (Chen et al., 2021). Given an offline dataset, we create three sub-datasets by filtering trajectories representing the top 5%, median 5%, and bottom 5% performance. We then leverage these sub-datasets to train three policies—referred to as *top*, *median*, and *bottom*—using behavior cloning. We then develop *TD3 with percentile behavior cloning (TD3+%BC)*, which optimizes the policy by $\max_\pi \mathbb{E}_{s\sim\mathcal{D}}[v^\pi(s) - \lambda(\pi(s) - \pi_\%(s))]$ where $\pi_\% \in \{\text{top, median, bottom}\}$. This variant of TD3+BC simply replaces the behavior policy with the percentile cloning policy. We refer to $\pi_\%$ as the *reference policy*. Our main hypothesis is that a better reference policy (e.g. top) can dramatically improve the efficiency of behavior regularization compared to a worse reference policy (e.g. bottom). Fig. 1 provides the results on *hopper-medium-replay*, *hopper-medium-expert*, *walker-medium-replay* and *walker-medium-expert* from the D4RL datasets (Fu et al., 2020). A few key observations. *First*, the efficacy of behavior regularization is strongly contingent on the reference policy. For example, TD3+top%BC dominates TD3+bottom%BC across all benchmarks. This confirms our hypothesis. *Second*, behavior regularization guarantees improvement. No matter using top, median or bottom, TD3+%BC produces a better or similar policy compared to the reference policy, clearly demonstrating the advantage of behavior regularization over behavior cloning.

Inspired by these findings, we ask: *is it possible to automatically discover good reference policies from the data to increase the efficiency of behavior regularization?* This paper attempts to give an affirmative answer to this question. Our key observation is that Conservative Policy Iteration, an algorithm that has been widely used for online RL (Abdolmaleki et al., 2018; Abbasi-Yadkori et al., 2019; Mei et al., 2019; Geist et al., 2019), can also be extended to the offline setting with minimal changes. The core concept behind this approach hinges on the iterative refinement of the reference policy utilized for behavior regularization. This iterative process enables the algorithm to implicitly avoid resorting to out-of-sample actions, all while ensuring continuous policy enhancement. We provide both theoretical and empirical evidence to establish that, for the tabular setting, our approach is capable of learning the optimal policy covered by the offline dataset, commonly referred to as the in-sample optimal policy (Kostrikov et al., 2022; Xiao et al., 2023). We then discuss several implementation details of the algorithm when function approximations are applied. Our method can be seamlessly implemented with just a few lines of code modifications built upon TD3+BC (Fujimoto & Gu, 2021). We evaluate our method on the D4RL benchmark (Fu et al., 2020). Experiment results show that it outperform previous state-of-the-art methods on the majority of tasks, with both rapid training speed and reduced computational overhead.

## 2 PRELIMINARIES

### 2.1 MARKOV DECISION PROCESS

We consider Markov Decision Process (MDP) determined by $M = \{\mathcal{S}, \mathcal{A}, P, r, \gamma\}$ (Puterman, 2014), where $\mathcal{S}$ and $\mathcal{A}$ represent the state and action spaces. The discount factor is given by $\gamma \in [0, 1)$, $r : \mathcal{S} \times \mathcal{A} \to \mathbb{R}$ denotes the reward function, $P : \mathcal{S} \times \mathcal{A} \to \Delta(\mathcal{S})$ defines the transition dynamics [1].

---

[1]We use $\Delta(\mathcal{X})$ to denote the set of probability distributions over $\mathcal{X}$ for a finite set $\mathcal{X}$.

Given a policy $\pi : \mathcal{S} \to \Delta(\mathcal{A})$, we use $\mathbb{E}^\pi$ to denote the expectation under the distribution induced by the interconnection of $\pi$ and the environment. The *value function* specifies the future discounted total reward obtained by following policy $\pi$,

$$V^\pi(s) = \mathbb{E}^\pi \left[ \sum_{t=0}^\infty \gamma^t r(s_t, a_t) \Big| s_0 = s \right] , \tag{1}$$

The *state-action value function* is defined as

$$Q^\pi(s, a) = r(s, a) + \gamma \mathbb{E}_{s' \sim P(\cdot|s,a)}[V^\pi(s')] . \tag{2}$$

There exists an *optimal policy* $\pi^*$ that maximizes values for all states $s \in \mathcal{S}$. The optimal value functions, $V^*$ and $Q^*$, satisfy the *Bellman optimality equation*,

$$V^*(s) = \max_a r(s, a) + \gamma \mathbb{E}_{s'}[V^*(s')], \ \ Q^*(s, a) = r(s, a) + \gamma \mathbb{E}_{s'} \left[ \max_{a'} Q^*(s', a') \right] . \tag{3}$$

## 2.2 Offline Reinforcement Learning

In this work, we consider learning an optimal decision making policy from previously collected offline dataset, denoted as $\mathcal{D} = \{s_i, a_i, r_i, s'_i\}_{i=0}^{n-1}$. The dataset is generated following this procedure: $s_i \sim \rho, a_i \sim \pi_{\mathcal{D}}, s'_i \sim P(\cdot|s_i, a_i), r_i = r(s_i, a_i)$, where $\rho$ represents an unknown probability distribution over states, and $\pi_{\mathcal{D}}$ is an *unknown behavior policy*. In offline RL, the learning algorithm can only take samples from $\mathcal{D}$ without collecting new data through interactions with the environment.

Behavior regularization is a simple yet efficient technique for offline RL (Kumar et al., 2019; Wu et al., 2019). It imposes a constraint on the learned policy to emulate $\pi_{\mathcal{D}}$ according to some distance measure. A popular choice is to use the KL-divergence (Fujimoto & Gu, 2021; Brandfonbrener et al., 2021)[2],

$$\max_\pi \mathbb{E}_{a \sim \pi} [Q(s, a)] - \tau D_{\mathrm{KL}} (\pi(s)||\pi_{\mathcal{D}}(s)) , \tag{4}$$

where $\tau > 0$ is a hyper-parameter. Here, $Q$ is some value function. Typical choices include the value of $\pi_{\mathcal{D}}$ (Brandfonbrener et al., 2021), or the value of the learned policy $\pi$ (Wu et al., 2019; Fujimoto & Gu, 2021). As shown by Fig. 1, the main limitation of behavior regularization is that it relies on the dataset being generated by an expert or near-optimal $\pi_{\mathcal{D}}$. When used on datasets derived from more suboptimal policies—typical of those prevalent in real-world applications—these methods do not yield satisfactory results.

## 3 Iteratively Refined Behavior Regularization

In this paper, we introduce a new offline RL algorithm by exploring idea of iteratively improving the reference policy used for behavior regularization. Our primary goal is to improve the robustness of existing behavior regularization methods while making minimal changes to existing implementation. We first introduce *conservative policy optimization (CPO)*, a commonly used technique in online RL, then describe how it can also shed some light on developing offline RL algorithms.

Let $\tau > 0$, the CPO update the policy for any $s \in \mathcal{S}$ with the following policy optimization problem

$$\max_\pi \mathbb{E}_{a \sim \pi} \left[ Q^{\bar\pi}(s, a) \right] - \tau D_{\mathrm{KL}} (\pi(s)||\bar\pi(s)) , \tag{5}$$

which generalizes behavior regularization (4) using an arbitrary *reference policy* $\bar\pi$. The idea is to optimize a policy without moving too far away from the reference policy to increase learning stability. As shown in the following proposition, this conservative policy update rule enjoys two intriguing properties: *first*, it guarantees policy improvement over the reference policy $\bar\pi$; and *second*, the updated policy still stays on the support of the reference policy. These properties also echo our empirical findings presented in Fig. 1: as a special case of CPO with $\bar\pi = \pi_{\mathcal{D}}$, behavior regularization is stable in offline learning and always produces a better policy than $\pi_{\mathcal{D}}$.

---

[2]We note that although Fujimoto & Gu (2021) applies a behavior cloning term as regularization, this is indeed a KL regularization under Gaussian parameterization with standard deviation.

**Proposition 1.** *let $\bar{\pi}^*$ be the optimal policy of (5). For any $s \in \mathcal{S}$, we have that $V^{\bar{\pi}^*}(s) \geq V^{\bar{\pi}}(s)$ ; and $\bar{\pi}^*(a|s) = 0$ given $\bar{\pi}(a|s) = 0$.*

*Proof.* The proof of this result, as well as other results, are provided in the Appendix. □

In summary, the conservative policy update (5) *implicitly guarantees policy improvement constrained on the support of the reference policy $\bar{\pi}$.* By extending this key observation in an iteratively manner, we obtain the following *Conservative Policy Iteration (CPI)* algorithm for offline RL. It starts with the behavior policy $\pi_0 = \pi_{\mathcal{D}}$. Then in each iteration $t = 0, 1, 2, \ldots$, the following computations are done:

- *Policy evaluation*: compute $Q^{\pi_t}$ and;
- *Policy improvement*: $\forall s \in \mathcal{S}, \pi_{t+1} = \arg\max_\pi \mathbb{E}_{a \sim \pi} [Q^{\pi_t}(s, a)] - \tau D_{\mathrm{KL}} (\pi || \pi_t)$ .

In this approach, the algorithm commences with the behavior policy and proceeds to iteratively refine the reference policy used for behavior regularization. Thanks to the conservative policy update, CPI ensures policy improvement while mitigating the risk of querying any OOD actions that could potentially introduce instability to the learning process.

We note that CPO is a special case of *mirror descent* in the online learning literature (Hazan et al., 2016). Extending this technique to sequential decision making has been previously investigated in online RL (Abdolmaleki et al., 2018; Abbasi-Yadkori et al., 2019; Mei et al., 2019; Geist et al., 2019). In particular, the *Politex* algorithm considers exactly the same update as (5) for online RL (Abbasi-Yadkori et al., 2019). This algorithm can be viewed as a softened or averaged version of policy iteration. Such averaging reduces noise of the value estimator and increases the robustness of policy update. Perhaps surprisingly, our key contribution is to show this simple yet powerful policy update rule also facilitates offline learning, as it guarantees policy improvement while implicitly avoid querying OOD actions.

### 3.1 THEORETICAL ANALYSIS

We now analyze the convergence properties of CPI. In particular, we consider the tabular setting with finite state and action space. Our analysis reveals that in this setting, CPI converges to the optimal policy that are well-covered by the dataset, commonly referred to as the in-sample optimal policy. Consider the *in-sample Bellman optimality equation* (Fujimoto et al., 2019)

$$V_{\pi_{\mathcal{D}}}^*(s) = \max_{a : \pi_{\mathcal{D}}(a|s) > 0} \left\{ r(s, a) + \gamma \mathbb{E}_{s' \sim P(\cdot|s,a)} \left[ V_{\pi_{\mathcal{D}}}^*(s') \right] \right\} . \tag{6}$$

This equation explicitly avoids bootstrapping from OOD actions while still guaranteeing optimality for transitions that are well-supported by the data. A fundamental challenge lies in the development of scalable algorithms for its resolution (Fujimoto et al., 2019; Kostrikov et al., 2022; Xiao et al., 2023). Our next result shows that CPI provides a simple yet effective solution.

**Theorem 1.** *We consider tabular MDPs with finite $\mathcal{S}$ and $\mathcal{A}$. Let $\pi_t$ be the produced policy of CPI at iteration $t$. There exists a parameter $\tau > 0$ such that for any $s \in \mathcal{S}$*

$$V_{\pi_{\mathcal{D}}}^*(s) - V^{\pi_t}(s) \leq \frac{1}{(1-\gamma)^2} \sqrt{\frac{2 \log |\mathcal{A}|}{t}} . \tag{7}$$

To ensure robustness to data distribution, an offline RL algorithm must possess the *stitching* capability, which involves seamlessly integrating suboptimal trajectories from the dataset. In-sample optimality provides a formal definition to characterize this ability. As discussed in previous works and confirmed by our experiments, existing behavior regularization approaches, such as TD3+BC (Fujimoto & Gu, 2021), fall short in this regard. In contrast, Theorem 1 suggests that iteratively refining the reference policy for behavior regularization can enable the algorithm to acquire the stitching ability, marking a significant improvement over existing approaches. Our empirical studies in the experiments section further support this claim.

---

**Algorithm 1** CPI & CPI-RE

---

1: Initialize two actors networks $\pi_1$, $\pi_2$ and critic networks $q_1$, $q_2$ with random parameters $\omega^1$, $\omega^2$, $\theta^1$, $\theta^2$; target networks $\bar{\theta}^1 \longleftarrow \theta^1$, $\bar{\theta}^2 \longleftarrow \theta^2$, $\bar{\omega}^1 \longleftarrow \omega^1$, $\bar{\omega}^2 \longleftarrow \omega^2$.
2: **for** $t = 0, 1, 2..., T$ **do**
3:     Sample a mini-batch of transitions $(s, a, r, s')$ from offline dataset $\mathcal{D}$
4:     Update the parameters of critic $\theta^i$ using equation 8
5:     **if** $t$ mod 2 **then**
6:         // For CPI
7:         Copy the historical snapshot of $\omega^1$ to $\omega^2$ and update $\omega^1$ using equation 10
8:         // For CPI-RE
9:         Choose the current best policy between $\omega^1$ and $\omega^2$ as the reference policy
10:         Update $\omega^1$ and $\omega^2$ using equation 10 in a sequential manner
11:         Update target networks
12:     **end if**
13: **end for**

---

## 4 PRACTICAL IMPLEMENTATIONS

In this section we discuss how to implement CPI properly when function approximation is applied. Throughout this section we develop algorithms for continuous actions. Extension to discrete action setting is straightforward. We build CPI as an actor-critic algorithm based on TD3+BC (Fujimoto & Gu, 2021). We learn an actor $\pi_\omega$ with parameters $\omega$, and critic $Q_\theta$ with parameters $\theta$. The policy $\pi_\omega$ is parameterized using a Gaussian policy with learnable mean (Fujimoto et al., 2018). We also normalize features of every states in the offline dataset as discussed in (Fujimoto & Gu, 2021).

CPI consists of two steps at each iteration. The first step involves policy evaluation, which is carried out using standard TD learning: we learn the critic by

$$\min_\theta \mathbb{E}_{s,a,r,s'\sim\mathcal{D},a'\sim\pi_\omega(s')} \left[ \frac{1}{2} \left( r + \gamma Q_{\bar{\theta}}(s', a') - Q_\theta(s, a) \right)^2 \right] , \tag{8}$$

where $Q_{\bar{\theta}}$ is a target network. We also apply the double-Q trick to stabilize training (Fujimoto et al., 2018; Fujimoto & Gu, 2021). The policy improvement step requires more careful algorithmic design. The straightforward implementation is to use gradient descent on the following

$$\max_{\omega'} \mathbb{E}_{s\sim\mathcal{D}} \left[ \mathbb{E}_{a\sim\pi_{\omega'}}[Q_\theta(s, a)] - \tau D_{\mathrm{KL}}(\pi_\omega(s)||\bar{\pi}(s)) \right] . \tag{9}$$

Here, the reference policy $\bar{\pi}$ is a copy of the current parameters $\omega$ and kept frozen during optimization. This leads to the so-called *iterative actor-critic* or *multi-step* algorithm (Brandfonbrener et al., 2021). In their study, Brandfonbrener et al. (2021) observe that this iterative algorithm frequently encounters practical challenges, mainly attributed to the substantial variance associated with off-policy evaluation. Similar observations have also been made in our experiments (See Fig. 4). We conjecture that while Proposition 1 establishes that the exact solution of (9) remains within the data support when the actor is initialized as $\pi_\omega = \pi_\mathcal{D}$, practical implementations often rely on a limited number of gradient descent steps for optimizing (9). This leads to policy optimization errors which are further exacerbated iteratively. To overcome this issue, we find it is useful to also add the original behavior regularization,

$$\max_{\omega'} \mathbb{E}_{s\sim\mathcal{D}} \left[ \mathbb{E}_{a\sim\pi_{\omega'}}[Q_\theta(s, a)] - \tau\lambda D_{\mathrm{KL}}(\pi_\omega(s)||\bar{\pi}(s)) - \tau(1-\lambda)D_{\mathrm{KL}}(\pi_\omega(s)||\pi_\mathcal{D}(s)) \right] , \tag{10}$$

which further constrains the policy on the support of data to enhance learning stability. The parameter $\lambda$ balances between one-step policy improvement and behavior regularization. Note that adding the term $D_{\mathrm{KL}}(\pi_\omega(s)||\bar{\pi}(s))$ is the only difference compared to TD3+BC. In practice this can be done with one line code modification based on TD3+BC implementations.

**Ensembles of Reference Policy** One limitation of (10) lies in the potential for a negligible difference between the learning policy and the reference policy, due to the limited gradient steps when optimizing. This minor discrepancy may restrict the policy improvement over the reference policy. To improve the efficiency of policy improvement, we explore the idea of using an ensembles of reference policies. In particular, we apply two policies with independently initialized parameters $\omega^1$ and $\omega^2$. Let $Q_{\theta^1}$

and $Q_{\theta^2}$ be the value functions of these two policies respectively. When updating the parameters $\omega^i$ for $i \in \{1, 2\}$, we choose the current best policy as the reference policy, where the superiority is decided according to the current value estimate. In other words, we only enable a superior reference policy to elevate the performance of the learning policy, preventing the learning policy from being dragged down by an inferior reference policy. We call this algorithm *Conservative Policy iteration with Reference Ensembles (CPI-RE)*. We give the pseudocode of both CPI and CPI-RE in Algorithm 1. For CPI, the training process is exactly the same with that of TD3+BC except for the policy updating (marked in orange). The difference between CPI-RE and CPI is also reflected in the policy updating (marked in blue).

## 5 EXPERIMENT

We subject our algorithm to a series of rigorous experimental evaluations. We first present an empirical study to exemplify CPI's optimality in the tabular setting. Then, we compare the practical implementations of CPI, utilizing function approximation, against prior state-of-the-art algorithms in the D4RL benchmark tasks (Fu et al., 2020), to highlight its superior performance. In addition, we also present the resource consumption associated with different algorithms. Finally, comprehensive analysis of various designs to scrutinize their impact on the algorithm's performance are provided.

### 5.1 OPTIMALITY IN THE TABULAR SETTING

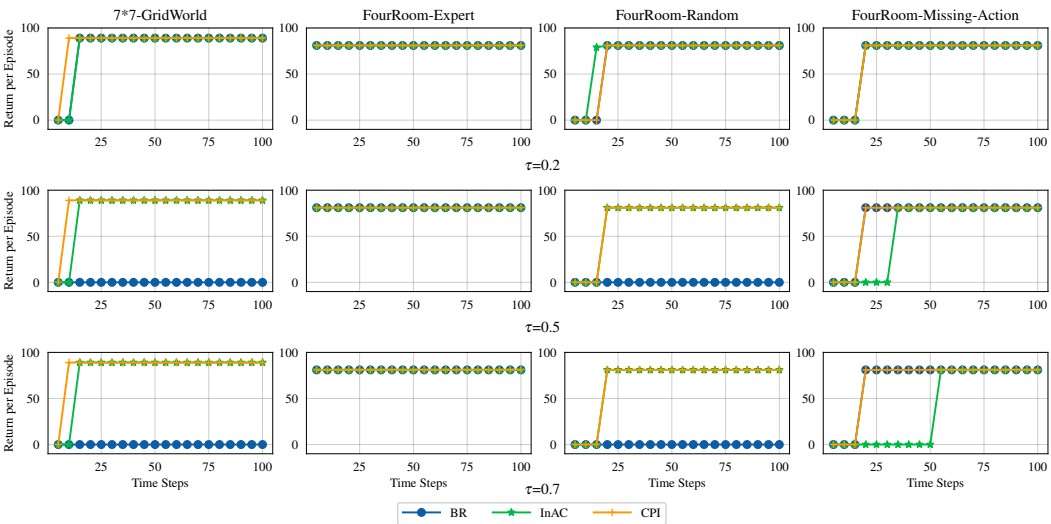

Figure 2: Training curves of BR, InAC and CPI on 7*7-GridWorld and FourRoom. CPI converges to the oracle across various $\tau$ and environments settings, similar to the in-sample optimal policy InAC.

We first conduct an evaluation of CPI within two GridWorld environments. The first environment's map comprises a $7 * 7$ grid layout and the second environment's map is the FourRoom (Xiao et al., 2023). In both environments, the agent is tasked with navigating from the bottom-left to the goal positioned in the upper-right in as few steps as possible. The agent has access to four actions: *{up, down, right, left}*. The reward is set to -1 for each movement, with a substantial reward of 100 upon reaching the goal; this incentivizes the agent to minimize the number of steps taken. Each episode is terminated after 30 steps, and $\gamma$ is set to 0.9. We use an *inferior behavior policy* to collect 10k transitions, of which the action probability is *{up:0.1, down:0.4, right:0.1, left:0.4}* at every state in the $7 * 7$ grid environment. For the FourRoom environment, we use three types of behavior policy to collect data: (1) Expert dataset: collect 10k transitions with the optimal policy; (2) Random dataset: collect 10k transitions with a random restart and equal probability of taking each action; (3) Missing-Action dataset: remove all *down* actions in transitions of the upper-left room from the Mixed dataset. Although some behavior policies are suboptimal, the optimal path is ensured to exist in the offline data, in which case a clear algorithm should still be able to identify the optimal path.

We consider two baseline algorithms: InAC (Xiao et al., 2023), a method that guarantees to find the in-sample softmax, and a method that employs policy iteration with behavior regularization (BR), which could be viewed as an extension of TD3+BC (Fujimoto & Gu, 2021) for the discrete action setting. The policies derived from each method are evaluated using greedy strategy that selects actions with highest probability.

As illustrated in Fig. 2, both CPI and InAC converge to the oracle across various $\tau$ and environments settings. In contrast, BR underperforms when a larger $\tau$ is applied on 7*7-GridWorld and FourRoom-random, as larger $\tau$ means more powerful constraint on the learning policy, BR thus become more similar to the behavioral policy.

## 5.2 RESULTS ON CONTINUOUS CONTROL PROBLEMS

Table 1: Average normalized scores of CPI with the mean and standard deviation and previous methods on the D4RL benchmark. D-QL is short for Diffusion-QL. CPI achieves best overall performance among all the methods and consumes quite few computing resources. The top-3 results on each dataset is marked as bold.

| Dataset | DT | TD3+BC | CQL | IQL | POR | EDAC | InAC | D-QL | CPI | CPI-RE |
|---|---|---|---|---|---|---|---|---|---|---|
| halfcheetah-random | 2.2 | 11.0 | **31.3** | 13.7 | 29.0 | 28.4 | 19.6 | 22.0 | **29.7±1.1** | **30.7±0.4** |
| hopper-random | 5.4 | 8.5 | 5.3 | 8.4 | 12.0 | 25.3 | **32.4** | 18.3 | **29.5±3.7** | **30.4±2.9** |
| waker2d-random | 2.2 | 1.6 | 5.4 | 5.9 | **6.3** | **16.6** | 6.3 | 5.5 | **5.9±1.7** | 5.5±0.9 |
| halfcheetah-medium | 42.6 | 48.3 | 46.9 | 47.4 | 48.8 | **65.9** | 48.3 | 51.5 | **64.4±1.3** | **65.9±1.6** |
| hopper-medium | 67.6 | 59.3 | 61.9 | 66.3 | **98.2** | **101.6** | 60.3 | 96.6 | **98.5±3.0** | 97.9±4.4 |
| waker2d-medium | 74.0 | 83.7 | 79.5 | 78.3 | 81.1 | **92.5** | 82.7 | 87.3 | 85.8±0.8 | **86.3±1.0** |
| halfcheetah-medium-replay | 36.6 | 44.6 | 45.3 | 44.2 | 43.5 | **61.3** | 44.3 | 48.3 | **54.6±1.3** | **55.9±1.5** |
| hopper-medium-replay | 82.7 | 60.9 | 86.3 | 94.7 | 98.9 | 101.0 | 92.1 | 102.0 | **101.7±1.6** | **103.2±1.4** |
| waker2d-medium-replay | 66.6 | 81.8 | 76.8 | 73.9 | 76.6 | 87.1 | 69.8 | 98.0 | **91.8±2.9** | **93.8±2.2** |
| halfcheetah-medium-expert | 86.8 | 90.7 | 95.0 | 86.7 | 94.7 | **106.3** | 83.5 | 97.2 | 94.7±1.1 | **95.6±0.9** |
| hopper-medium-expert | 107.6 | 98.0 | 96.9 | 91.5 | 90.0 | **110.7** | 93.8 | **112.3** | 106.4±4.3 | **110.1±4.1** |
| waker2d-medium-expert | 108.1 | 110.1 | 109.1 | 109.6 | 109.1 | **114.7** | 109.0 | **111.2** | 110.9±0.4 | **111.2±0.5** |
| halfcheetah-expert | 87.7 | 96.7 | **97.3** | 94.9 | 93.2 | **106.8** | 93.6 | 96.3 | 96.5±0.2 | **97.4±0.4** |
| hopper-expert | 94.2 | 107.8 | 106.5 | 108.8 | **110.4** | 110.1 | 103.4 | 102.6 | **112.2±0.5** | **112.3±0.5** |
| walker2d-expert | 108.3 | 110.2 | 109.3 | 109.7 | 102.9 | **115.1** | 110.6 | 109.5 | **110.6±0.1** | **111.2±0.2** |
| Gym-MuJoCo Total | 972.6 | 1013.2 | 1052.8 | 1034.0 | 1094.7 | **1243.4** | 1049.7 | 1158.6 | **1193.2** | **1207.4** |
| antmaze-umaze | 59.2 | 78.6 | 74.0 | 87.5 | 76.8 | 16.7 | 84.8 | **96.0** | **98.8±1.1** | **99.2±0.5** |
| antmaze-umaze-diverse | 53.0 | 71.4 | 84.0 | 62.2 | 64.8 | 0.0 | 82.4 | 84.0 | **88.6±5.7** | **92.6±10.0** |
| antmaze-medium-play | 0.0 | 3.0 | 61.2 | 71.2 | **87.2** | 0.0 | - | 79.8 | **82.4±5.8** | **84.8±5.0** |
| antmaze-medium-diverse | 0.0 | 10.6 | 53.7 | 70.0 | 75.2 | 0.0 | - | 82.0 | **80.4±8.9** | **80.6±11.3** |
| antmaze-large-play | 0.0 | 0.0 | 15.8 | **39.6** | 24.4 | 0.0 | - | **49.0** | 20.6±16.3 | **33.6±8.1** |
| antmaze-large-diverse | 0.0 | 0.2 | 14.9 | 47.5 | **59.2** | 0.0 | - | **61.7** | 45.2±6.9 | **48.0±6.2** |
| Antmaze Total | 112.2 | 163.8 | 303.6 | 378.0 | 387.6 | 16.7 | - | **452.5** | **416.0** | **438.8** |
| pen-human | 73.9 | -1.9 | 35.2 | 71.5 | **76.9** | 52.1 | 52.3 | 75.7 | **80.1±16.9** | **87.0±25.3** |
| pen-cloned | 67.3 | 9.6 | 27.2 | 37.3 | 67.6 | **68.2** | -8.0 | 60.8 | **71.8±35.2** | **70.7±15.8** |
| Adroit Total | 141.2 | 7.7 | 62.4 | 108.8 | **144.5** | 120.3 | 44.3 | 136.5 | **151.9** | **157.7** |
| Total | 1226.0 | 1184.7 | 1418.8 | 1520.8 | **1626.8** | 1380.4 | - | **1747.6** | **1761.2** | **1803.9** |
| Runtime (s/epoch) | - | **7.4** | - | - | - | 19.6 | - | 39.8 | **8.5** | 19.1 |
| GPU Memory (GB) | - | **1.4** | - | - | - | 1.9 | - | 1.5 | **1.4** | **1.4** |

In this section we provide a suite of results using three continuous control tasks from D4RL (Fu et al., 2020): Mujoco, Antmaze, and Adroit. Mujoco, a benchmark often used in previous studies forms the basis of our experimental framework. Adroit is a high-dimensional robotic manipulation task with sparse rewards. Antmaze, with its sparse reward property, necessitates that the agent learns to discern segments within sub-optimal trajectories and to assemble them, thereby discovering the complete trajectory leading to a rewardable position. Given that the majority of datasets for these tasks contain a substantial volume of sub-optimal or low-quality data, relying solely on behavior-regularization may be detrimental to performance.

We compare CPI with several baselines, including DT (Chen et al., 2021), TD3+BC (Fujimoto & Gu, 2021), CQL (Kumar et al., 2020), IQL (Kostrikov et al., 2022), POR (Xu et al., 2022), EDAC (An et al., 2021), Diffusion-QL (Wang et al., 2022), and InAC (Xiao et al., 2023). The results of baseline methods are either reproduced by executing the official code or sourced directly from the original papers. Unless otherwise specified, results are depicted with 95% confidence intervals, represented by shaded areas in figures and expressed as standard deviations in tables. The average normalized

results of the final 10 evaluations for Mujoco and Adroit, and the final 100 evaluations for Antmaze, are reported. More details of the experiments are provided in the Appendix.

Our experiment results, summarized in Table 1, clearly show CPI outperforms baselines in overall. In most Mujoco tasks, CPI surpasses the extant, widely-utilized algorithms, and it only slightly trails behind the state-of-the-art EDAC method. For Antmaze and Adroit, CPI's performance is on par with the top-performing methods such as POR and Diffusion-QL. We also evaluate the resource consumption of different algorithms from two aspects: (1) runtime per training epoch (1000 gradient steps); (2) GPU memory consumption. The results in Table 1 show that CPI requires fewer resources which could be beneficial for practitioners.

### 5.3 ABLATION STUDIES

#### 5.3.1 EFFECT OF REFERENCE ENSEMBLE

We provide learning curves of CPI and CPI-RE on Antmaze in Fig. 3 to further show the efficacy of using ensemble of reference policies. CPI-RE exhibits a more stable performance compared to vanilla CPI, also outperforming IQL. Learning curves on other domains are provided in the Appendix.

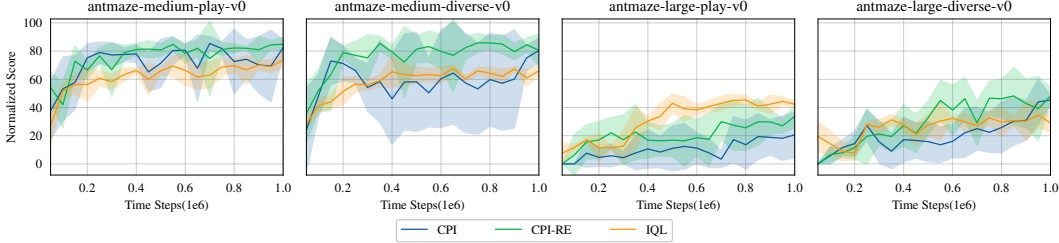

Figure 3: The learning curves of CPI, CPI-RE, and IQL on Antmaze. With reference ensemble, CPI-RE exhibits a more stable performance compared to vanilla CPI, also outperforming IQL.

#### 5.3.2 EFFECT OF USING BEHAVIOR REGULARIZATION

A direct implementation based on the theoretical results observed in the tabular setting could be derived. Specifically, we initialize $\pi_\omega = \pi_\mathcal{D}$ and execute the CPI without incorporating behavior regularization. For each gradient step, we perform *one-step* or *multi-step* updates for both policy evaluation and policy improvement. The empirical outcomes presented in Fig.4 underscore the poor performance of this straightforward approach. Similar trends are also observed in (Brandfonbrener et al., 2021). Such phenomenon arises primarily because, in the presence of function approximation, the deviation emerges when policy optimization does not necessarily remain within the defined support without the behavior regularization.

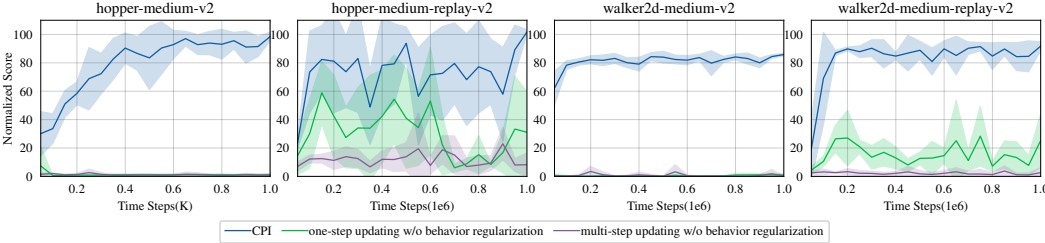

Figure 4: Effect of using Behavior Regularization. Without it, both one-step and multi-step updating methods at each gradient step suffer from deviation derived from out-of-support optimization.

#### 5.3.3 EFFECT OF USING DIFFERENT KL STRATEGIES

In our implementation of the CPI method, we employ the reverse KL divergence. This is distinct from the forward KL divergence approach adopted by (Nair et al., 2020). A comprehensive ablation of

incorporating different KL divergence strategies in CPI is presented in Fig.5. As evidenced from the results, our reverse KL-based CPI exhibits superior performance compared to the forward KL-based implementations. Implementations details of CPI with forward KL are provided in Appendix.

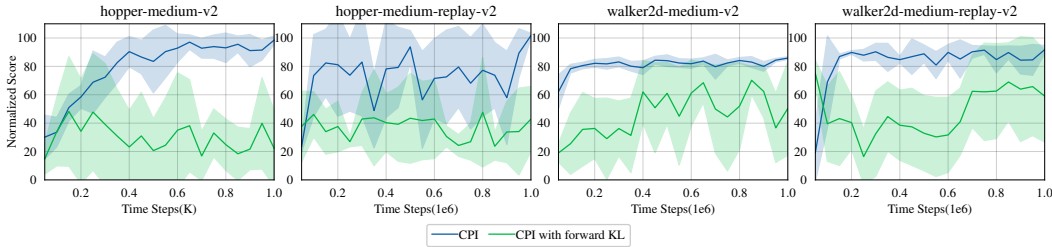

Figure 5: Effect of using different KL strategies. Reverse KL-based CPI exhibits superior performance compared to the forward KL-based CPI. See Appendix for details of forward KL.

### 5.3.4 EFFECT OF HYPER PARAMETERS

Fig. 6 illustrates the effects of using different hyperparameters $\tau$ and $\lambda$, offering valuable insights for algorithm tuning. The weighting coefficient $\lambda$ regulates the extent of behavior policy integration into the training and affects the training process. As shown in Fig. 6a, when $\lambda = 0.1$, the early-stage performance excels, as the behavior policy assists in locating appropriate actions in the dataset. However, this results in suboptimal final convergence performance, attributable to the excessive behavior policy constraint on performance improvement. For larger values, such as 0.9, the marginal weight of the behavior policy leads to performance increase during training. Unfortunately, the final performance might be poor. This is due to that the policy does not have sufficient behavior cloning guidance, leading to a potential distribution shift during the training process. Consequently, we predominantly select a $\lambda$ value of 0.5 or 0.7 to strike a balance between the reference policy regularization and behavior regularization. The regularization parameter $\tau$ plays a crucial role in determining the weightage of the joint regularization relative to the Q-value component. We find that (Fig. 6b) $\tau$ assigned to dataset of higher quality and lower diversity (e.g., expert dataset) ought to be larger than those associated with datasets of lower quality and higher diversity (e.g., medium dataset).

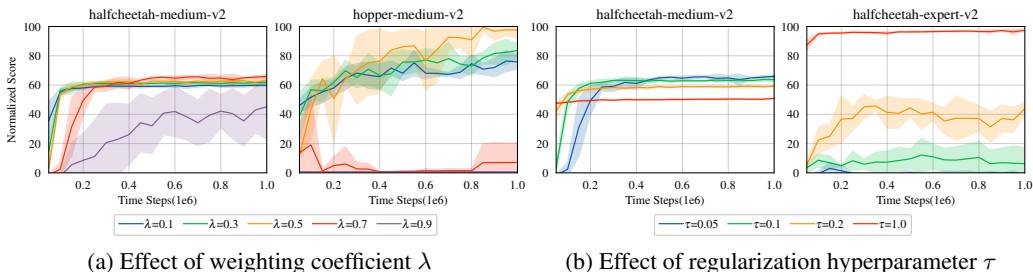

(a) Effect of weighting coefficient $\lambda$        (b) Effect of regularization hyperparameter $\tau$

Figure 6: Hyperparameters ablation studies. The tuning experience drawn from this include: (1) $\lambda$ could mostly be set to 0.5 or 0.7; (2) The higher the dataset quality is, the higher $\tau$ should be set.

## 6 CONCLUSION

In this paper, we propose an innovative offline RL algorithm termed *Conservative Policy iteration (CPI)*. By iteratively refining the policy used for behavior regularization, CPI progressively improves itself within the behavior policy's support and provably converges to the in-sample optimal policy in the tabular setting. We then propose practical implementations of CPI for tackling continuous control tasks. Experimental results on the D4RL benchmark show that two practical implementations of CPI surpass previous cutting-edge methods in a majority of tasks of various domains, offering both expedited training speed and diminished computational overhead. Nonetheless, our study is not devoid of limitations. For instance, our method's performance with function approximation is

contingent upon the selection of two hyperparameters, which may necessitate tuning for optimal results. Future research may include exploring CPI's potential in resolving offline-to-online tasks by properly relaxing the support constraint during online fine-tuning.

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

# A   PROOFS

We first introduce some technical lemmas that will be used in the proof.

We consider a $k$-armed one-step decision making problem. Let $\Delta$ be a $k$-dimensional simplex and $\boldsymbol{q} = (q(1), \dots, q(k)) \in \mathbb{R}^k$ be the reward vector. Maximum entropy optimization considers

$$\max_{\pi \in \Delta} \pi \cdot \boldsymbol{q} + \tau \mathbb{H}(\pi). \tag{11}$$

The next result characterizes the solution of this problem (Lemma 4 of (Nachum et al., 2017)).

**Lemma 1.** *For $\tau > 0$, let*

$$F_\tau(\boldsymbol{q}) = \tau \log \sum_a e^{q(a)/\tau}, \quad f_\tau(\boldsymbol{q}) = \frac{e^{\boldsymbol{q}/\tau}}{\sum_a e^{q(a)/\tau}} = e^{\frac{\boldsymbol{q} - F_\tau(\boldsymbol{q})}{\tau}}. \tag{12}$$

*Then there is*

$$F_\tau(\boldsymbol{q}) = \max_{\pi \in \Delta} \pi \cdot \boldsymbol{q} + \tau \mathbb{H}(\pi) = f_\tau(\boldsymbol{q}) \cdot \boldsymbol{q} + \tau \mathbb{H}(f_\tau(\boldsymbol{q})). \tag{13}$$

The second result gives the error decomposition of applying the Politex algorithm to compute an optimal policy. This result is adopted from (Abbasi-Yadkori et al., 2019).

**Lemma 2.** *Let $\pi_0$ be the uniform policy and consider running the following iterative algorithm on a MDP for $t \geq 0$,*

$$\pi_{t+1}(a|s) \propto \pi_t(a|s) \exp\left(\frac{q^{\pi_t}(a|s)}{\tau}\right), \tag{14}$$

*Then*

$$v^*(s) - v^{\pi_t}(s) \leq \frac{1}{(1-\gamma)^2} \sqrt{\frac{2 \log |\mathcal{A}|}{t}}. \tag{15}$$

*Proof.* We use vector and matrix operations to simply the proof. In particular, we use $v^\pi \in \mathbb{R}^{|\mathcal{S}|}$ and $q^\pi \in \mathbb{R}^{|\mathcal{S}| \times |\mathcal{A}|}$. Let $P \in \mathbb{R}^{|\mathcal{S}||\mathcal{A}| \times |\mathcal{S}|}$ be the transition matrix, and $P^\pi \in \mathbb{R}^{|\mathcal{S}| \times |\mathcal{S}|}$ be the transition matrix between states when applying the policy $\pi$.

We first apply the following error decomposition

$$v^{\pi^*} - v^{\pi_{k-1}} = v^{\pi^*} - \frac{1}{k} \sum_{i=0}^{k-1} v^{\pi_i} + \frac{1}{k} \sum_{i=0}^{k-1} v^{\pi_i} - v^{\pi_{k-1}}. \tag{16}$$

For the first part,

$$v^{\pi^*} - \frac{1}{k} \sum_{i=0}^{k-1} v^{\pi_i} \tag{17}$$

$$= \frac{1}{k} \sum_{i=0}^{k-1} (I - \gamma P^{\pi^*})^{-1} (T^{\pi^*} v^{\pi_i} - v^{\pi_i}) \tag{18}$$

$$= \frac{1}{k} \sum_{i=0}^{k-1} (I - \gamma P^{\pi^*})^{-1} (M^{\pi^*} - M^{\pi_i}) q^{\pi_i} \tag{19}$$

$$\leq \frac{1}{(1-\gamma)^2} \sqrt{\frac{2 \log |\mathcal{A}|}{k}}, \tag{20}$$

where Eq. (18) follows by the value difference lemma, Eq. (20) follows by applying the regret bound of mirror descent algorithm for policy optimization. For the second part,

$$\frac{1}{k}\sum_{i=0}^{k-1} v^{\pi_i} - v^{\pi_{k-1}} \tag{21}$$

$$= \frac{1}{k}\sum_{i=0}^{k-1}(I - \gamma P^{\pi_k-1})^{-1}(v^{\pi_i} - T^{\pi_{k-1}} v^{\pi_i}) \tag{22}$$

$$= \frac{1}{k}\sum_{i=0}^{k-1}(I - \gamma P^{\pi_k-1})^{-1}(M^{\pi_i} - M^{\pi_{k-1}})q^{\pi_i} \tag{23}$$

$$\leq 0 \tag{24}$$

where for the last step we use that for any $s \in \mathcal{S}$, $\sum_{i=0}^{k-1}(\pi_i - \pi_{k-1})\hat{q}_i \leq 0$. This follows by

$$\sum_{i=0}^{k-1} \pi_{k-1}\hat{q}_i = \sum_{i=0}^{k-2} \pi_{k-1}\hat{q}_i + \pi_{k-1}\hat{q}_{k-1} + \tau\mathcal{H}(\pi_{k-1}) - \tau\mathcal{H}(\pi_{k-1}) \tag{25}$$

$$\geq \sum_{i=0}^{k-2} \pi_{k-2}\hat{q}_i + \pi_{k-1}\hat{q}_{k-1} + \tau\mathcal{H}(\pi_{k-2}) - \tau\mathcal{H}(\pi_{k-1}) \tag{26}$$

$$\geq \cdots\cdots \tag{27}$$

$$\geq \sum_{i=0}^{k-1} \pi_i\hat{q}_i + \tau\mathcal{H}(\pi_0) - \tau\mathcal{H}(\pi_{k-1}) \tag{28}$$

$$\geq \sum_{i=0}^{k-1} \pi_i\hat{q}_i , \tag{29}$$

where Eq. (26) follows by applying Lemma 1 and the definition of $\pi_{k-1}$, Eq. (29) follows by the definition of $\pi_0$. Combine the above together finishes the proof. □

*Proof of Theorem 1.* First recall the in-sample optimality equation

$$q^*_{\pi_{\mathcal{D}}}(s,a) = r(s,a) + \gamma\mathbb{E}_{s'\sim P(\cdot|s,a)}\left[\max_{a':\pi_{\mathcal{D}}(a'|s')>0} q^*_{\pi_{\mathcal{D}}}(s',a')\right] , \tag{30}$$

which could be viewed as the optimal value of a MDP $M_{\mathcal{D}}$ covered by the behavior policy $\pi_{\mathcal{D}}$, where $M_{\mathcal{D}}$ only contains transitions starting with $(s,a) \in \mathcal{S} \times \mathcal{A}$ such that $\pi_{\mathcal{D}}(a|s) > 0$. Then the result can be proved by two steps. First, note that the CPI algorithm will never consider actions such that $\pi_{\mathcal{D}}(a|s) = 0$. This is directly implied by Lemma 1. Second, we apply Lemma 2 to show the error bound of using CPI on $M_{\mathcal{D}}$. This finishes the proof.

□

**Proposition 2.** *let $\bar{\pi}^*$ be the optimal policy of (34). For any $s \in \mathcal{S}$, we have that $V^{\bar{\pi}^*}(s) \geq V^{\bar{\pi}}(s)$ ; and $\bar{\pi}^*(a|s) = 0$ given $\bar{\pi}(a|s) = 0$.*

*Proof of Proposition 1.* We first prove the first part.

$$\mathbb{E}_{a\sim\bar{\pi}^*}\left[Q^{\bar{\pi}}(s,a)\right] \geq \mathbb{E}_{a\sim\bar{\pi}^*}\left[Q^{\bar{\pi}}(s,a)\right] - \tau D_{\mathrm{KL}}\left(\bar{\pi}^*(s)||\bar{\pi}(s)\right) \tag{31}$$

$$\geq \mathbb{E}_{a\sim\bar{\pi}}\left[Q^{\bar{\pi}}(s,a)\right] - \tau D_{\mathrm{KL}}\left(\bar{\pi}(s)||\bar{\pi}(s)\right) \tag{32}$$

$$= \mathbb{E}_{a\sim\bar{\pi}}\left[Q^{\bar{\pi}}(s,a)\right] , \tag{33}$$

where the first inequality follows the non-negativity of KL divergence, the second inequality follows $\bar{\pi}^*$ is the optimal policy of (34). Then the first statement can be proved by applying a standard recursive argument. The second statement is directly implied by Lemma 1.

$$\max_\pi \mathbb{E}_{a\sim\pi}\left[Q^{\bar{\pi}}(s,a)\right] - \tau D_{\mathrm{KL}}\left(\pi(s)||\bar{\pi}(s)\right) , \tag{34}$$

□

## B    DETAILED EXPERIMENTAL SETTINGS

In this section we provide the complete details of the experiments in our paper.

### B.1    THE EFFECT OF DIFFERENT REGULARIZATIONS

We analyze the impact on the algorithm when different policies are used as regularization terms. The most straightforward way to obtain policies with different performances is the baseline method $X\%$BC mentioned in DT (Chen et al., 2021). We set $x$ to 5 in order to make the difference between policies more significant and let the 5%BC policy be the policy used for constraint. In detail, we selectively choose different 5% data for behavioral cloning so that we can get various policies with different performances. First, we sort the trajectories in the dataset by their return (accumulated rewards) and select three different levels of data: top (highest return), middle or bottom (lowest return). Each

Table 2: 5% BC Hyperparameters

| Hyperparameter | Value |
|---|---|
| Hidden layers | 3 |
| Hidden dim | 256 |
| Activation function | ReLU |
| Mini-batch size | 256 |
| Optimizer | Adam |
| Dropout | 0.1 |
| Learning rate | 3e-4 |

level of data is sampled at 5% of the total data volume. Then we can train 5%BC using the MLP network via BC and get three 5%BC policies with different performances. We can then use each 5%BC policy as the regularization term of TD3 to implement the TD3+5%BC method. Besides, we also normalize the states and set the regularization parameter $\alpha$ to 2.5. The only difference between the TD3+5%BC and TD3+BC is the action obtained in the regularization term. In TD3+BC, the action used for constraint is directly obtained from the dataset according to the corresponding state in the dataset, while in TD3+5%BC, the action for constraint is sampled using 5%BC. We set the training steps for 5% BC to 5e5 and the training steps for TD3+5%BC to 1M. Table 2 concludes the hyperparameters of 5% BC. The hyperparameters of TD3+5%BC are the same as those of TD3+BC (Fujimoto & Gu, 2021).

### B.2    BASELINES

We conduct experiments on the benchmark of D4RL and use Gym-MuJoCo datasets of version v2, Antmaze datasets of version v0, and Adroit datasets of version v1. We compare CPI with BC, DT (Chen et al., 2021), TD3+BC (Fujimoto & Gu, 2021), CQL (Kumar et al., 2020), IQL (Kostrikov et al., 2022), POR (Xu et al., 2022), EDAC (An et al., 2021), Diffusion-QL (Wang et al., 2022) and InAC (Xiao et al., 2023). In Gym-Mujoco tasks, our experimental results are preferentially selected from EDAC, Diffusion-QL papers, or their original papers. If corresponding results are unavailable from these sources, we rerun the code provided by the authors. Specifically, we run it on the expert dataset for POR[3]. For DT[4] and IQL[5], we follow the hyperparameters given by the authors to run on random and expert datasets. For Diffusion-QL[6], we set the hyperparameters for the random dataset to be the same as on the medium-replay dataset and the hyperparameters for the expert dataset to be the same as on the medium-expert dataset according to the similarity between these datasets. For InAC[7], we run it on the random dataset. In Antmaze tasks, our experimental results are taken from the Diffusion-QL paper, except for EDAC[8] and POR. The results of EDAC are obtained by running the authors' provided code and setting the Q ensemble number to 10 and $\eta = 1.0$. Even when we transformed the rewards according to (Kumar et al., 2020), the performance of EDAC on Antmaze still did not perform well, which matches the report in the offline reinforcement learning library CORL (Tarasov et al., 2022). As the results in the POR paper are under Antmaze v2, we rerun them under Antmaze v0. In the Adroit task, we rerun the experiment for TD3+BC, DT (with return-to-go set to 3200), POR (with the same parameters as the antmaze tasks), and InAC(with tau set to 0.7).

---

[3]https://github.com/ryanxhr/POR
[4]https://github.com/kzl/decision-transformer
[5]https://github.com/ikostrikov/implicit_policy_improvement
[6]https://github.com/Zhendong-Wang/Diffusion-Policies-for-Offline-RL
[7]https://github.com/hwang-ua/inac_pytorch
[8]https://github.com/snu-mllab/EDAC

Table 3: CPI Hyperparameters

|  | Hyperparameter | Value |
| --- | --- | --- |
| Architecture | Actor hidden layers | 3 |
|  | Actor hidden dim | 256 |
|  | Actor activation function | ReLU |
|  | Critic hidden layers | 3 |
|  | Critic hidden dim | 256 |
|  | Critic activation function | ReLU |
| Learning | Optimizer | Adam |
|  | Critic learning rate | 3e-4 for MuJoCo and Adroit |
|  |  | 1e-3 for Antmaze |
|  | Actor learning rate | 3e-4 |
|  | Mini-batch size | 256 |
|  | Discount factor | 0.99 for MuJoCo and Adroit |
|  |  | 0.995 for Antmaze |
|  | Target update rate | 5e-3 |
|  | Policy noise | 0.2 |
|  | Policy noise clipping | (-0.5, 0.5) |
|  | $\tau$ | {0.05, 0.2, 1, 2} for MuJoCo |
|  |  | {0.03, 0.05, 0.1, 1} for Antmaze |
|  |  | {100, 200} for Adroit |
|  | $\lambda$ | {0.5, 0.7} |

## B.3  CONSERVATIVE POLICY ITERATION

To implement our idea, we made slight modifications to TD3+BC[9] to obtain CPI. For CPI, we select a historical snapshot policy as the reference policy. Specifically, the policy snapshot $\pi^{k-2}$, which is two gradient steps before the current step, is chosen as the reference policy for the current learning policy $\pi^k$. CPI is trained similarly to TD3. For CPI-RE, the complete network contains two identical policy networks with different initial parameters, so that two policies with distinct performances can be obtained. The two policy networks in CPI-RE are updated via cross-update to fully utilize the information from both value networks. During training, the value network evaluates actions induced by the two policy networks, and only the higher value action is used to pull up the performance of the learning policy. During evaluation, the two policy networks are also used to select high-value actions to interact with the environment. CPI only has one more actor compared to TD3+BC and therefore requires less computational overhead than other state-of-the-art offline RL algorithms with complex algorithmic architectures, such as EDAC (an ensemble of Q functions) and Diffusion-QL (Diffusion model).

According to TD3+BC, we normalize the state of the MuJoCo tasks and use the original rewards in the dataset. For the Antmaze datasets, we ignore the state normalization techniques and transform the rewards in the dataset according to (Kumar et al., 2020). For Adroit datasets, we also do not use state normalization and standardize the rewards according to Diffusion-QL (Wang et al., 2022). To get the reported results, we average the returns of 10 trajectories with five random seeds evaluated every 5e3 steps for MuJoCo and Adroit, 100 trajectories with five random seeds evaluated every 5e4 steps for Antmaze. In addition, we evaluate the runtime and the memory consumption of different algorithms to train an epoch (1000 gradient update steps). All experiments are run on a GeForce GTX 2080TI GPU.

The most critical hyperparameters in CPI are the weight coefficient $\lambda$ and the regularization parameter $\tau$. On all datasets, the choice of $\lambda = 0.5$ or $\lambda = 0.7$ is the most appropriate so that the two actions (from the behavioral policy $\beta$ and the reference policy $\bar{\pi}$) can be well-weighed to participate in the learning process. As mentioned in the main text, the choice of $\tau$ depends heavily on the characteristics of the dataset. For a high-quality dataset, $\tau$ should be larger to learn in a close imitation way, and for a high-diversity dataset, the $\tau$ should be chosen to be smaller to make the whole learning process more similar to RL. We find that training is more stable and better performance can be achieved on Antmaze when the critic learning rate is set to 1e-3. Also, since Antmaze is a sparse reward domain,

---
[9]https://github.com/sfujim/TD3_BC

Table 4: Regularization parameter $\tau$ and weighting factor $\lambda$ of CPI for all datasets. On most datasets, CPI and CPI-RE have the same optimal parameters. On some datasets, the values in parentheses indicate the parameters of CPI, and the values outside the parentheses indicate the parameters of CPI-RE.

| Dataset | regularization parameter $\tau$ | weighting coefficient $\lambda$ |
|---|---|---|
| halfcheetah-random-v2 | 0.05 | 0.7 |
| halfcheetah-medium-v2 | 0.05 | 0.7 |
| halfcheetah-medium-replay-v2 | 0.05 | 0.7 |
| halfcheetah-medium-expert-v2 | 2 | 0.5 |
| halfcheetah-expert-v2 | 1 | 0.5 |
| hopper-random-v2 | 0.05 | 0.5 |
| hopper-medium-v2 | 0.2 | 0.5 |
| hopper-medium-replay-v2 | 0.05 | 0.5 |
| hopper-medium-expert-v2 | 1 | 0.5(0.7) |
| hopper-expert-v2 | 1 | 0.5(0.7) |
| walker2d-random-v2 | 2 | 0.5 |
| walker2d-medium-v2 | 1 | 0.7 |
| walker2d-medium-replay-v2 | 0.2 | 0.7(0.5) |
| walker2d-medium-expert-v2 | 1 | 0.7 |
| walker2d-expert-v2 | 1 | 0.7 |
| antmaze-umaze-v0 | 1 | 0.7 |
| antmaze-umaze-diverse-v0 | 0.1 | 0.5 |
| antmaze-medium-play-v0 | 0.1 | 0.5 |
| antmaze-medium-diverse-v0 | 0.1(0.05) | 0.5 |
| antmaze-large-play-v0 | 0.03 | 0.5(0.7) |
| antmaze-large-diverse-v0 | 0.05 | 0.5 |
| pen-human-v1 | 100(200) | 0.5(0.7) |
| pen-cloned-v1 | 100 | 0.7 |

we also set the discount factor to 0.995. Table 4 gives our selections of hyperparameters $\tau$ and $\lambda$ on different datasets. Other settings are given in Table 3.

## C CPI WITH FORWARD KL

We show how to optimize (34) using the *forward KL*. Extension to (9) is straightforward by picking $\bar{\pi} = \pi_t$.

Recall (34),

$$\max_{\pi} \mathbb{E}_{a \sim \pi} \left[ Q^{\bar{\pi}}(s, a) \right] - \tau D_{\mathrm{KL}} \left( \pi(s) \| \bar{\pi}(s) \right) . \tag{35}$$

By Lemma 1, the optimal policy $\bar{\pi}^*$ has a closed form solution.

$$\bar{\pi}^*(a|s) \propto \bar{\pi}(a|s) \exp \left( \frac{Q^{\bar{\pi}}(s, a)}{\tau} \right) . \tag{36}$$

This implies that to optimize (34), we can also consider the following

$$\min_{\pi} D_{\mathrm{KL}}(\bar{\pi}^* \| \pi) \propto -\bar{\pi}^* \log \pi \tag{37}$$

$$= \mathbb{E}_{a \sim \bar{\pi}^*} [\log \pi(a|s)] \tag{38}$$

$$= \mathbb{E}_{a \sim \bar{\pi}} \left[ \frac{\bar{\pi}^*(a|s)}{\bar{\pi}(a|s)} \log \pi(a|s) \right] \tag{39}$$

$$= \mathbb{E}_{a \sim \bar{\pi}} \left[ \frac{\exp(Q^{\bar{\pi}}(s, a)/\tau)}{Z(s)} \log \pi(a|s) \right] . \tag{40}$$

The first step follows by removing terms not dependent on $\pi$. $Z(s) = \sum_a \bar{\pi}(a|s) \exp(Q^{\bar{\pi}}(s, a))$ is a normalization term. In practice, it is often approximated by a state value function (Xiao et al., 2023; Kostrikov et al., 2022; Nair et al., 2020).

**Connection to (34)** We now discuss how to connect the forward KL objective described above with the original optimization problem (34). Indeed, it can be verified that (34) corresponds to a *reverse KL* policy optimization,

$$\arg\min_{\pi} D_{\mathrm{KL}}(\pi||\bar{\pi}^*) = \arg\min_{\pi} \pi \log \pi - \pi \log \bar{\pi}^* \tag{41}$$

$$= \arg\min_{\pi} \pi \log \pi - \pi \log \frac{\bar{\pi} \exp\left(Q^{\bar{\pi}}/\tau\right)}{Z} \tag{42}$$

$$= \arg\min_{\pi} \pi \log \pi - \pi \log \bar{\pi} - \pi Q^{\bar{\pi}}/\tau + \log Z \tag{43}$$

$$= \arg\min_{\pi} D_{\mathrm{KL}}(\pi||\bar{\pi}) - \pi Q^{\bar{\pi}}/\tau \tag{44}$$

$$= \arg\max_{\pi} \pi Q^{\bar{\pi}} - \tau D_{\mathrm{KL}}(\pi||\bar{\pi}). \tag{45}$$

where we use vector notations for simplicity. Although forward and reverse KL has exactly the same solution in the tabular case, when function approximation is applied these two objectives can showcase different optimization properties. We refer to Chan et al. (2022) for more discussions on these two objectives.

## D    COMPARISON OF TD3+BC AND CPI

According to the experimental part of the ablation study and the analysis in the (Wu et al., 2022), $\alpha$ is an important parameter for controlling the constraint strength. TD3+BC is set $\alpha$ to a constant value of 2.5 for each dataset, whereas CPI chooses the appropriate $\tau$ from a set of $\tau$ alternatives. We note that the hyperparameter that plays a role in regulating Q and regularization in CPI is $\tau$, which can essentially be understood as the reciprocal of $\alpha$ in TD3+BC. Therefore, for the convenience of comparison, we rationalize the reciprocal of $\tau$ as the parameter $\alpha$. In this section, we set the $\alpha$ of TD3+BC to be consistent with that of CPI in order to show that the performance improvement of CPI mainly comes from amalgamating the benefits of both behavior-regularized and in-sample algorithms. Further, we also compare CPI with TD3+BC with dynamically changed $\alpha$ (Beeson & Montana, 2022), which improves TD3+BC by a large margin, to show the superiority of CPI. The selection of parameters is shown in Table 4.

The results for TD3+BC (vanilla), TD3+BC (same $\alpha$ with CPI), TD3+BC with dynamically changed $\alpha$ and CPI are shown in Table 5. Comparing the variants of TD3+BC with different $\alpha$ choices, it can be found that changing $\alpha$ can indeed improve the performance of TD3+BC. However, compared with TD3+BC (same $\alpha$), the performance of CPI is significantly better, which proves the effectiveness of the mechanism for iterative refinement of policy for behavior regularization in CPI.

Table 5: Comparison with TD3+BC and its variants.

| Dataset | TD3+BC (vanilla) | TD3+BC (same $\alpha$ with CPI) | TD3+BC (dynamic $\alpha$) | CPI | CPI-RE |
|---|---|---|---|---|---|
| halfcheetah-medium | 48.3 | 58.8 | 55.3 | 64.4 | 65.9 |
| hopper-medium | 59.3 | 69.0 | 100.1 | 98.5 | 97.9 |
| waker2d-medium | 83.7 | 79.8 | 89.1 | 85.8 | 86.3 |
| halfcheetah-medium-replay | 44.6 | 51.2 | 48.7 | 54.6 | 55.9 |
| hopper-medium-replay | 60.9 | 88.5 | 100.5 | 101.7 | 103.2 |
| waker2d-medium-replay | 81.8 | 83.0 | 87.9 | 91.8 | 93.8 |
| halfcheetah-medium-expert | 90.7 | 92.3 | 91.9 | 94.7 | 95.6 |
| hopper-medium-expert | 98.0 | 76.1 | 103.9 | 106.4 | 110.1 |
| waker2d-medium-expert | 110.1 | 109.4 | 112.7 | 110.9 | 111.2 |
| halfcheetah-expert | 96.7 | 94.5 | 97.5 | 96.5 | 97.4 |
| hopper-expert | 107.8 | 110.8 | 112.4 | 112.2 | 112.3 |
| walker2d-expert | 110.2 | 109.3 | 113.0 | 110.6 | 111.2 |
| Above Total | 992.1 | 1022.7 | 1113.0 | 1128.1 | 1140.8 |
| halfcheetah-random | 11.0 | 23.2 | - | 29.7 | 30.7 |
| hopper-random | 8.5 | 28.8 | - | 29.5 | 30.4 |
| waker2d-random | 1.6 | 4.4 | - | 5.9 | 5.5 |
| Gym Total | 1013.2 | 1079.1 | - | 1193.3 | 1207.4 |
| antmaze-umaze | 78.6 | 93.8 | - | 98.8 | 99.2 |
| antmaze-umaze-diverse | 71.4 | 73.2 | - | 88.6 | 92.6 |
| antmaze-medium-play | 3.0 | 13.0 | - | 82.4 | 84.8 |
| antmaze-medium-diverse | 10.6 | 8.0 | - | 80.4 | 80.6 |
| antmaze-large-play | 0.0 | 0.0 | - | 20.6 | 33.6 |
| antmaze-large-diverse | 0.2 | 0.0 | - | 45.2 | 48.0 |
| Antmaze Total | 163.8 | 188.0 | - | 416.0 | 438.8 |

# E MORE EXPERIMENTAL RESULTS

## E.1 EFFECT OF DIFFERENT NUMBERS OF REFERENCE POLICIES

We also present the effect of using different numbers of actors in CPI-RE. We can conclude from Figure 7 that increasing the actor number from 1 to 2 (i.e., introducing the reference policy) can significantly improve the performance of the learning policy. As further introducing reference policy derived from more actors does not bring significant benefits and entails significant resource consumption. Thus, we set the actor number in our method to two.

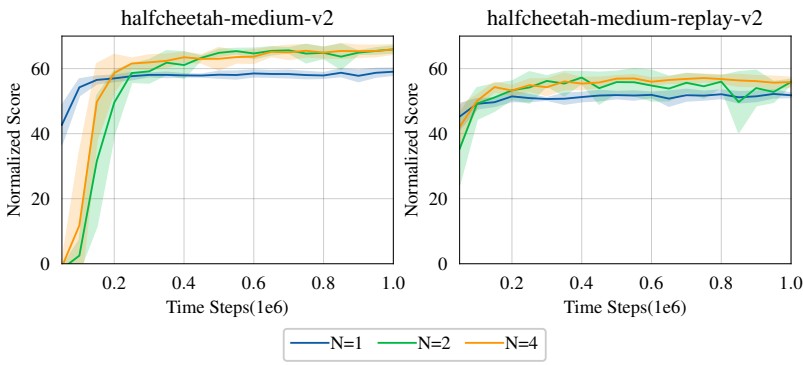

Figure 7: Effect of different actor number

## E.2 LEARNING CURVES OF CPI

The learning curve of CPI on MuJoCo tasks is shown in Figure 8.

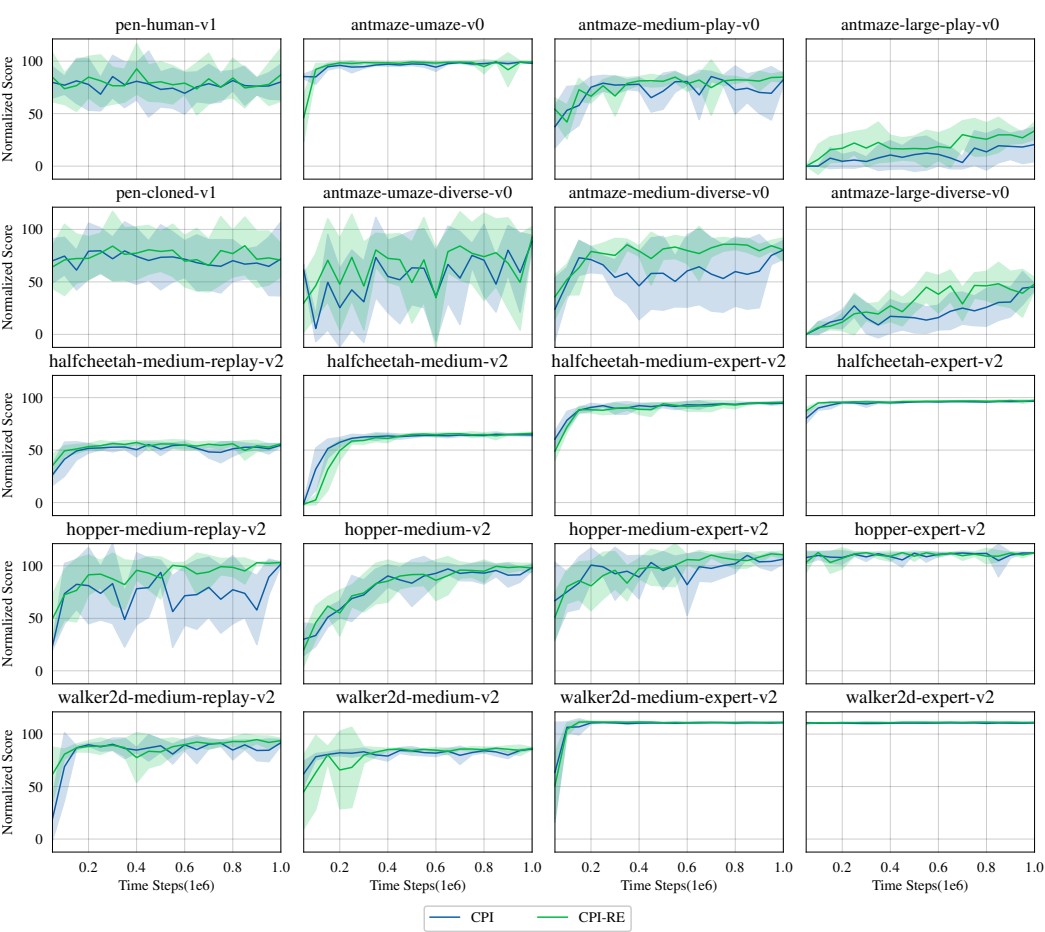

Figure 8: Learning curves of CPI and CPI-RE for Mujoco, Antmaze and Pen, evaluated every 5e3 steps.

