# OpenReview forum: "Iteratively Refined Behavior Regularization for Offline Reinforcement Learning"
_ICLR.cc/2024/Conference — Submitted to ICLR 2024_

### Official Review · Reviewer_ByMR · 2023-10-15

**Soundness:** 3 good
**Presentation:** 4 excellent
**Contribution:** 2 fair
**Rating:** 6
**Confidence:** 3

**Summary:**

This paper aims to address the suboptimality that arises from the behavior regularization w.r.t the suboptimal datasets in policy-constraint-based offline RL methods. Specifically, this paper casts policy constraints on an iteratively evolved reference policy $\bar{\pi}$ rather than the suboptimal behavior policy $\pi_D$. Theoretical analysis proves that this iteratively evolved policy constraint can not only avoid OOD queries but also achieves in-sample optimality when the initial reference policy $\bar{\pi}$ is initialized to $\pi_D$ under the full-data-coverage assumption that the optimzied policy $\pi$ should always stay in the support of the reference policy $\bar{\pi}$. However, for practical implementation, this paper also applies constraints on the suboptimal behavior policy to stablize training, where the constraint strength between $\pi_D$ and $\bar{\pi}$ are balanced by an newly introduced hyper-parameter. This paper also provides extensive experimental results to demonstrate the effectiveness of the proposed method.

**Strengths:**

1. This paper is well organized and well written, presenting good motivations in Figure 1 and theoretical analysis in Theorem 1 to show the significance of the proposed method.
2. The proposed method is simple and easy to implement.
3. The experimental results are sufficient.

**Weaknesses:**

## Major weakness
The performance improvement seems to primarily stem from comprehensive parameter tuning, rather than the proposed  iterative refinement of policy constraints. CPI can be viewed as adding an additional iteratively refined policy constraint to TD3+BC, while introducing a new hyperparameter $\lambda$ to adjust the constraint strengths. In my view, this is doing some kind of conservatism relaxation and introduces an additional hyperparameter, providing a more precise adjustment of conservatism strengths than solely tuning the $\tau$. Consequently, this leads to performance gains compared to the base method, TD3+BC.

See from Table 3 and Table 4 that hyper-parameters are thoroughly tuned for each individual task. Moreover, Table 5 shows that TD3+BC (dynamic $\alpha$) is on par with CPI and CPI-RE. This indicates that TD3+BC might also achieve in-sample optimality and meanwhile prevents OOD issues with a carefully sweeped conservatism strength $\alpha$.

I would consider raise my score if the major weakness is well resolved.

## Minor weakness
The idea about utilizing an iteratively refined policy constraint is already introduced by a recent offline2online RL paper[1], which is stated as a future research in Conclusion.

[1] PROTO: Iterative Policy Regularized Offline-to-Online Reinforcement Learning. 2023.

## Typo
1. In proposition 1, the (34) would be better (5).

**Questions:**

1. I'm wondering how do CPI and CPI-RE perform using one group of hyperparameters for each types of tasks including Gym-Mujoco, Antmaze and Adroit, respectively, just like previous works such as IQL and TD3+BC does.
2. In Equation (10), I'm questioning the term $\tau(1-\lambda)\lambda$. Could there possibly be an error? Should it be $\tau(1-\lambda)$ instead?

Please refer to the weakness for details.

---

> ### Author Response · Authors · 2023-11-14
> **Response to Reviewer ByMR**
>
> ## Q: The performance improvement seems to primarily stem from comprehensive parameter tuning, rather than the proposed iterative refinement of policy constraints.
>
> A: In Section 5.1, we conduct an evaluation of CPI within two GridWorld environments. We consider two baseline algorithms: InAC, a method that guarantees to find the in-sample softmax, and a method that employs policy iteration with behavior regularization (BR), which could be viewed as an extension of TD3+BC  for the discrete action setting.  Both CPI and InAC converge to the oracle across various $\tau$ and environments settings. In contrast, BR underperforms when a larger $\tau$ is applied on 7*7-GridWorld and FourRoom-random. This evaluation ensures no parameter tuning while still demonstrates the effectiveness of CPI. , which proves the  performance improvement primarily stem from the  proposed iterative refinement of policy constraints.
>
> While when function approximation is applied, tuning parameters becomes a common issue of almost all SOTA offline RL algorithms. For example,  XQL[1], an in-sample optimization method, is relies on tuning the hyper-parameters according to Figure 7, 10 and 11 in the appendix of the original paper. In practice, we believe it's unlikely that a method could dominate all tasks without tuning parameters.
>
>
>
> ## Q: TD3+BC might also achieve in-sample optimality and meanwhile prevents OOD issues with a carefully swept conservatism strength $\alpha$.
>
> A: The critical hyper-parameter in TD3+BC is $\alpha$ value, which controls the weight of RL learning and behavior cloning process. The $\alpha$ is set 2.5 by default in the original paper in Mujoco tasks, and we indeed find that the value of 2.5 works well only in Mujoco tasks. Thus, as you suggested, we conduct a hyperparameter search in the ranges {0.0001, 0.05, 0.25, 2.5, 25, 36, 50, 100} for Antmaze and Adroit datasets, and report the results in our results tables.  It can be found that changing $\alpha$ can indeed improve the performance of TD3+BC. However, the performance of CPI is still significantly better on Antmaze and Adroit, which proves the effectiveness of the mechanism for iterative refinement of policy for behavior regularization in CPI.
>
>
>
> |                        | TD3BC (default) | TD3BC (with swept best hyperparameter ) | CPI               | CPI-RE            |
> | ---------------------- | --------------- | --------------------------------------- | ----------------- | ----------------- |
> | antmaze-umaze          | 78.6            | 78.6 （$\alpha=2.5$）                   | **98.8$\pm$1.1**  | **99.2$\pm$0.5**  |
> | antmaze-umaze-diverse  | 71.4            | 71.4 （$\alpha=2.5$）                   | **88.6$\pm$5.7**  | **92.6$\pm$10.0** |
> | antmaze-medium-play    | 3.0             | 35.7$\pm$11.0 （$\alpha=36$）           | **82.4$\pm$5.8**  | **84.8$\pm$5.0**  |
> | antmaze-medium-diverse | 10.6            | 17.5$\pm$0.43（$\alpha=25$）            | **80.4$\pm$8.9**  | **80.6$\pm$11.3** |
> | antmaze-large-play     | 0.0             | 0.0$\pm$0.0 （$\alpha=50$）             | **20.6$\pm$16.3** | **33.6$\pm$8.1**  |
> | antmaze-large-diverse  | 0.2             | 0.2 （$\alpha=2.5$）                    | **45.2$\pm$6.9**  | **48.0$\pm$6.2**  |
> | pen-cloned             | 5.13            | 64.2 $\pm$17.3（$\alpha=0.0001$）                | **71.8$\pm$35.2** | **70.7$\pm$15.8** |
>
>
> ## Q: how do CPI and CPI-RE perform using one group of hyperparameters for each types of tasks
>
> A: We are still working on the experiments. We will post the results when the experiments finish.
>
>
>
> ## Q: term $\tau(1-\lambda)$
>
> A: Thanks for pointing this out. We have modify this in the revised paper.
>
>
>
> ## Q: In proposition 1, the (34) would be better (5).
>
> A:  This is due to a replicated label used in the same equation within method and appendix. We have revised them.
>
> ## Citations
>
>  [1] Garg, Divyansh, et al. "Extreme Q-Learning: MaxEnt RL without Entropy." *The Eleventh International Conference on Learning Representations*. 2022.

---

> ### Comment · Reviewer_ByMR · 2023-11-16
> **Additional concerns**
>
> Dear authors,
> I appreciate the authors for their detailed responses. Some of my concerns are addressed. However, I still have further concerns.
>
> 1. Regarding this response,
> >In practice, we believe it's unlikely that a method could dominate all tasks without tuning parameters.
>
> As the authors proposed a relatively simple method, I believe this kind of simple method should be robust enough for different tasks using the same hyperparameters. Otherwise, the simplicity of the proposed method goes nowhere as we still require a lot of hyper-parameter tuning to obtain a good result. For example, TD3+BC/CQL/IQL/POR/EDAC in the paper all used as little as possible hyper-parameter tuning to ensure good results.  Therefore, based on the fact that CPI and CPI-RE thoroughly tuned the hyper-parameters, other baselines should also be well-tuned to ensure a fair comparison.
>
> 2. Why does Table 1 bold CPI and CPI-RE for some tasks that are inferior to IQL, such as some antmaze tasks?

---

> ### Author Response · Authors · 2023-11-17
>
> Thanks for your reply!
> ### Q: The performance of CPI and CPI-RE with as little as possible hyper-parameter tuning?
> A: We provide the well-tuned TD3BC in the previous response, and the results show that CPI is still significantly better than the well-tuned TD3BC on Antmaze and Adroit. In addition, we agree that a simple method could reach realtively satisfying performance with as little as possible hyper-parameter (hp) tuning. Therefore, as you suggested, we used as few hyperparameters as possible on the same domain and conducted experiments. For MuJoCO datasets, we set $\tau$ to 0.05 or 1.0, $
> \lambda$ to 0.7. For Antmaze, we  set $\tau$ to 0.1 and $\lambda$ to 0.5. For Antmaze, we  set $\tau$ to 200.0 and $\lambda$ to 0.7. The detailed results (average across five seeds) and setting are provided in the following table:
> | Dataset                   | $\tau$ | $\lambda$ | TD3+BC | CQL    | IQL    | CPI with minimal hp tuning        | CPI results in paper |
> | ------------------------- | ------ | -------- | ------ | ------ | ------ | ------------- | -------------------- |
> | halfcheetah-random        | 0.05   | 0.7      | 11.0   | 31.3   | 13.7   | 29.7$\pm$1.1  | 29.7$\pm$1.1         |
> | halfcheetah-medium        | 0.05   | 0.7      | 48.3   | 46.9   | 47.4   | 64.4$\pm$1.3  | 64.4$\pm$1.3         |
> | halfcheetah-medium-replay | 0.05   | 0.7      | 44.6   | 45.3   | 44.2   | 54.6$\pm$1.3  | 54.6$\pm$1.3         |
> | halfcheetah-medium-expert | 1.0    | 0.7      | 90.7   | 95.0   | 86.7   | 89.7$\pm$1.5  | 94.7$\pm$1.1         |
> | halfcheetah-expert        | 1.0    | 0.7      | 96.7   | 97.3   | 94.9   | 98.1$\pm$0.6  | 96.5$\pm$0.2         |
> | hopper-random             | 0.05   | 0.7      | 8.5    | 5.3    | 8.4    | 29.5$\pm$3.8  | 29.5$\pm$3.7         |
> | hopper-medium             | 1.0    | 0.7      | 59.3   | 61.9   | 66.3   | 61.4$\pm$1.9  | 98.5$\pm$3.0         |
> | hopper-medium-replay      | 0.05   | 0.7      | 60.9   | 86.3   | 94.7   | 99.3$\pm$6.8  | 101.7$\pm$1.6        |
> | hopper-medium-expert      | 1.0    | 0.7      | 98.0   | 96.9   | 91.5   | 106.4$\pm$4.3 | 106.4$\pm$4.3        |
> | hopper-expert             | 1.0    | 0.7      | 107.8  | 106.5  | 108.8  | 112.2$\pm$0.5 | 112.2$\pm$0.5        |
> | walker2d-random           | 1.0    | 0.7      | 1.6    | 5.4    | 5.9    | 3.9$\pm$2.2   | 5.9$\pm$1.7          |
> | walker2d-medium           | 1.0    | 0.7      | 83.7   | 79.5   | 78.3   | 85.8$\pm$0.8  | 85.8$\pm$0.8         |
> | walker2d-medium-replay    | 1.0    | 0.7      | 81.8   | 76.8   | 73.9   | 89.1$\pm$3.0  | 91.8$\pm$2.9         |
> | walker2d-medium-expert    | 1.0    | 0.7      | 110.1  | 109.1  | 109.6  | 110.9$\pm$0.4 | 110.9$\pm$0.4        |
> | walker2d-expert           | 1.0    | 0.7      | 110.2  | 109.3  | 109.7  | 110.6$\pm$0.1 | 110.6$\pm$0.1        |
> | mujoco-total              | -      | -        | 1013.2 | 1052.8 | 1034.0 | 1145.6        | 1193.2               |
> | antmaze-umaze             | 0.1    | 0.5      | 78.6   | 74.0   | 87.5   | 96.6$\pm$3.2  | 98.8$\pm$1.1         |
> | antmaze-umaze-diverse     | 0.1    | 0.5      | 71.4   | 84.0   | 62.2   | 88.6$\pm$5.7  | 88.6$\pm$5.7         |
> | antmaze-medium-play       | 0.1    | 0.5      | 3.0    | 61.1   | 71.2   | 82.4$\pm$5.8  | 82.4$\pm$5.8         |
> | antmaze-medium-diverse    | 0.1    | 0.5      | 10.6   | 53.7   | 70.0   | 78.8$\pm$8.9  | 80.4$\pm$8.9         |
> | antmaze-large-play        | 0.1    | 0.5      | 0.0    | 15.8   | 39.6   | 3.3$\pm$5.7   | 20.6$\pm$16.3        |
> | antmaze-large-diverse     | 0.1    | 0.5      | 0.2    | 14.9   | 47.5   | 23.0$\pm$4.3  | 45.2$\pm$6.9         |
> | antmaze-total             | -      | -        | 163.8  | 303.6  | 378.0  | 372.7         | 416.0                |
> | pen-human                 | 200.0  | 0.7      | -1.9   | 35.2   | 71.5   | 80.1$\pm$16.9 | 80.1$\pm$16.9        |
> | pen-cloned                | 200.0  | 0.7      | 9.6    | 27.2   | 37.3   | 69.7$\pm$17.0 | 71.8$\pm$35.2        |
> | adroit-total              | -      | -        | 7.7    | 62.4   | 108.8  | 149.8         | 151.9                |
>
> The results show that comapred with IQL, CQL and TD3BC. the overall performance of CPI with as little as possible hyper-parameter tuning is still significantlly better.
>
> ### Q: Why does Table 1 bold CPI and CPI-RE for some tasks that are inferior to IQL, such as some antmaze tasks?
> A: We are sorry we didn't claim the meaning of the bold. Our intention is to highlight the top results in the tabular. We have modified this part in the paper to make readers understand more clearly. Thanks for your advice!

---

> > ### Comment · Reviewer_ByMR · 2023-11-17
> > **Thanks for the quick response**
> >
> > Dear authors,
> >
> > I appreciate your quick responses, which address my major concern regarding the hyper-parameter tuning. Thus, I'm happy to increase my score to a 6.
> >
> > Some minor typo: does the $\alpha$ in your responses refer to the $\lambda$ in your paper?

---

> > > ### Author Response · Authors · 2023-11-17
> > >
> > > Thanks for raising score! And you're right, we have modified it to $\lambda$.

---

### Official Review · Reviewer_cTmM · 2023-10-27

**Soundness:** 3 good
**Presentation:** 3 good
**Contribution:** 2 fair
**Rating:** 3
**Confidence:** 4

**Summary:**

This paper presents an innovative approach to refining behavior-regularized offline RL algorithms, introducing the concept of Iterative Self Policy Improvement (ISPI) that progressively optimizes within the behavior policy's support. The authors also propose a practical implementation trick for ISPI, demonstrating its effectiveness against strong baselines across various tasks.

**Strengths:**

1. Introduces a novel and promising concept of refining behavior policy to address suboptimal behavior.
2. Section 3 adeptly explains the iterative solution and establishes a clear connection with online learning.
3. Comprehensive and robust experimental section, with comparisons against the latest SOTA methods, clearly highlighting the improvements over the TD3+BC backbone algorithm.

**Weaknesses:**

1. There's a noticeable discrepancy between the theoretical framework and the practical implementation. The theory is grounded on in-sample learning assumption, yet the Competitive Policy Improvement (CPI) paradigm isn't in line with this. The implementation appears to violate the in-sample property suggested by the theory.
2. This paper lacks a clear link to in-sample learning despite a strong motivation, potentially leading to reader confusion, especially due to the mismatch between theory and practice. A restructuring of the manuscript to improve clarity and coherence is recommended.

minors:
There are two hyperparameters in ISPI, which may be hard to tune in practice.

**Questions:**

n/a

---

> ### Author Response · Authors · 2023-11-14
> **Response to Reviewer cTmM**
>
> ## Q: There's a noticeable discrepancy between the theoretical framework and the practical implementation.
>
> A: The theoretical analysis considers the **tabular setting** with finite state and action space.  In this setting, CPI converges to the optimal policy that are well-covered by the dataset, commonly referred to as the in-sample optimal policy. However in practice, implementations often rely on a limited number of gradient descent steps for optimizing when function approximation is applied. To address this issue, we introduce another behavior regularization to ensure the stability of the implementation.
>
> The mismatch between theory and practice is only due to function approximation and using gradient descent as optimization tools. Such an issue cannot be avoided and also exists in previous algorithms. For example, to learn the in-sample max, IQL requires solving expectile regression [1] (similarly, XQL [2] requires solving gumble regression), but practical implementation of IQL also depends on a few steps of gradient descent. The in-sample softmax policy iteration introduced in InAC  also uses samples from the current policy in the policy evaluation step [3].
>
>
>
> ## Q:  There are two hyperparameters in ISPI, which may be hard to tune in practice.
>
> A: We provide the settings of regularization parameter $\tau$ and weighting factor $\lambda$ used in our main experiments in Table 4 of the appendix. The parameters  $\lambda$ are settled by sweeping in [0.1, 0.3, 0.5, 0.7, 0.9] and choosing the best performing one. For ease of tunning, we also provide an empirical suggestion in Section 5.3.4 for the potential algorithm practitioners.
>
>
> ## Citations
> [1] Kostrikov, Ilya, et al. "Offline Reinforcement Learning with Implicit Q-Learning." *The Tenth International Conference on Learning Representations*. 2021.
>
> [2] Garg, Divyansh, et al. "Extreme Q-Learning: MaxEnt RL without Entropy." *The Eleventh International Conference on Learning Representations*. 2022.
>
> [3] Xiao, Chenjun, et al. "The In-Sample Softmax for Offline Reinforcement Learning." *The Eleventh International Conference on Learning Representations*. 2022.

---

> > ### Comment · Reviewer_cTmM · 2023-11-17
> > **Official Response to In-sample Learning from Reviewer cTmM**
> >
> > Thanks for the reply, but I would like to keep my current rating because the in-sample optimality equation holds only when those actions are from the dataset  (IQL, POR, SQL (EQL), X-QL, IAC, InAC). Apperatenly, CPI does sample actions from $\pi_{w'}$.

---

> ### Author Response · Authors · 2023-11-17
> **Response to Reviewer cTmM**
>
> Thanks for your reply. It seems that the reviewer judge our contibution only based on terminology: "In-sample algorithms only uses samples from the data. CPI doesn't do this. Therefore in-sample optimality cannot hold for CPI. "
>
> In Section 3.1, we prove that (Theorem 1) in the tabular case, CPI converges to the in-sample optimal solution, defined by Eq.6. That means, CPI enjoys the same theoretical guarantees as BRAC, IQL and InAC in the tabular setting case. If the reviewer believes that this is not true, we kindly ask the reviewer to point out which step of our derivation is wrong.

---

> > ### Author Response · Authors · 2023-11-22
> >
> > Dear reviewer, we have provided detailed responses, but have not yet hear back from you. We will appreciate it deeply if you could reply our rebuttal. We are sincerely looking forward to further discussions to address the reviewers concerns to our best. Thanks!

---

> > > ### Comment · Reviewer_cTmM · 2023-11-23
> > > **Official Comments from Reviewer cTmM**
> > >
> > > Thank the authors for the prompt response!
> > >
> > > 1. First, the in-sample optimality equation does not hold in CPI in the practical algorithm, and thus all following proof/lemma/proposition based on the in-sample optimality equation do not hold. I understand that Theorem 1 holds in the tabular case, but this theory-practice gap is significant. I strongly suggest the authors re-organize its proof under the in-support optimization framework rather than in-sample optimization.
> > > 2. Experiments: I do not have questions related to experiments and I have also synchronized with the discussion with Reviewer ByMR.
> > >
> > > minors:
> > > wrong citation: BCQ does not have the in-sample optimality equation.

---

### Official Review · Reviewer_JFNQ · 2023-10-30

**Soundness:** 2 fair
**Presentation:** 2 fair
**Contribution:** 2 fair
**Rating:** 3
**Confidence:** 5

**Summary:**

The paper introduces an approach to improve behavior regularization through conservative policy iteration. This is achieved by incorporating an additional KL regularization term between the current policy and a static target policy network into the pre-existing TD3+BC objective function. The authors offer a theoretical analysis for the tabular case. They evaluate their proposed algorithm, CPI, as well as an ensemble policy variant, CPI-RE, using toy discrete datasets and the offline RL benchmark D4RL datasets.

**Strengths:**

The experiments conducted are comprehensive and of high quality.

**Weaknesses:**

1. The paper's clarity and organization could benefit from a thorough revision. Currently, the flow is disjointed, with readers needing to frequently navigate back and forth, e.g., Proposition 1 references an equation found in the appendix. Additionally, the algorithm box lacks clarity, which is elaborated upon in the questions section.

2. The paper's originality and novelty appear limited. The newly introduced component merely utilizes a target policy network for KL regularization—a technique already employed in algorithms like TRPO and PPO. My interpretation suggests that the paper's primary contribution is a trick that demonstrates how integrating a conservative policy update can enhance offline RL.

3. The scope of testing is narrow. Evaluating only on TD3+BC doesn't provide enough evidence to support the universal benefits of adding conservative policy learning. The authors claim that implementing their method requires minimal modifications to existing algorithms. If so, it would be beneficial to test it on other frameworks like IQL, CQL, and Diffusion-QL.

**Questions:**

1. In Figure 1, does the dashed line represent a BC trained solely on the curated dataset?

2. You mentioned, "We provide the pseudocode for both CPI and CPI-RE here." I assume this "here" refers to the algorithm box?

3. The algorithm box seems ambiguous in its description:
  * Right after Equation (9), the reference policy is identified as the target policy. However, in the algorithm for CPI-RE, the reference policy is described as the optimal policy between $\omega_1$ and $\omega_2$.
  * Can you clarify what is meant by the "cross-update scheme"?

4. The paper notes that $\pi_\omega$ is initialized as $\pi_\omega = \pi_D$. Could you specify where this initialization is reflected in the algorithm box?

---

> ### Author Response · Authors · 2023-11-14
> **Response to Reviewer JFNQ**
>
> ## Q: Proposition 1 references an equation found in the appendix
>
> A: This is due to a replicated label used in the same equation within method and appendix. We have revised them.
>
>
>
> ## Q:  the algorithm box lacks clarity
>
> A: We provide the pseudocode for both CPI and CPI-RE in Algorithm 1. For line 9, it's true that the reference policy is the optimal policy between $w_{1}$ and $w_{2}$. As in CPI-RE,  two policies are trained. For example, when $w_{1}$ is updated, we compare the Q-value of actions from $w_{1}$ and  $w_{2}$. If the action of $w_{1}$ has higher Q-value, we will choose $w_{1}$ as the current reference policy. In this situation, only the behavior cloning term plays a role, which avoids the better policy $w_{1}$ being dragged down by the inferior policy $w_{2}$. Else, $w_{2}$ will be chose as the current reference policy.
>
> We try to express the sequential updating of $w_{1}$ and $w_{2}$ (i.e., update $w_{1}$ first, then update $w_{2}$) using the word cross-updating, we have modified this to avoid misunderstanding. Thanks for your advice.
>
>
>
> ## Q: the paper's primary contribution is a trick
>
> A: While we appreciate the reviewer's comparison with TRPO and PPO, our approach offers a unique contribution. We don't merely employ a target policy network for KL regularization but derive it from the conservative policy updates in the tabular offline RL context. In other words,  the iterative refinement of policy constraints is the core idea of our work and adding the behavior cloning term is to help the implementation when function approximation is applied. This is not just a 'trick' but a significant enhancement to offline RL both in the theoretical aspects under the tabular settings. In the practical evaluation, we also outperform existing SOTA offline RL baselines.
>
>
>
> ## Q: Evaluating only on TD3+BC doesn't provide enough evidence to support the universal benefits of adding conservative policy learning.
>
> A: We add conservative policy learning on IQL and test the variant on the following datasets. For each dataset, the weight of conservative policy learning is set to 1. For other parameters, we follow the original settings in IQL. Below are the final average normalized scores across three seeds trained after1M gradient steps.
>
> |                           | IQL   | IQL with conservative policy learning |
> | ------------------------- | ----- | ------------------------------------- |
> | halfcheetah-medium        | 47.4  | 48.3 $\pm$ 0.3                        |
> | halfcheetah-medium-replay | 44.2  | 44.5 $\pm$ 0.5                        |
> | halfcheetah-medium-expert | 86.7  | **94.6 $\pm$ 0.1**                    |
> | hopper-medium             | 66.3  | 58.6 $\pm$ 6.6                        |
> | hopper-medium-replay      | 94.7  | **100.6 $\pm$ 0.4**                   |
> | hopper-medium-expert      | 91.5  | **112.3 $\pm$ 0.4**                   |
> | walker2d-medium           | 78.3  | 76.4 $\pm$ 3.4                        |
> | walker2d-medium-replay    | 73.9  | **84.3 $\pm$ 3.1**                    |
> | walker2d-medium-expert    | 109.6 | **111.2 $\pm$ 0.9**                   |
>
> Note that while it's efficient to add conservative policy learning on IQL, it may depart from the original intention of our algorithm design. This is because in CPI and CPI-RE, the behavior cloning term is used to assist the iterative refinement of policy constraints to ensure the the learning policy staying in the dataset distribution.  While in the implementation in IQL with conservative policy learning, the iterative refinement of policy constraints becomes an auxiliary loss.
>
>
>
> ## Q: In Figure 1, does the dashed line represent a BC trained solely on the curated dataset?
>
> A: The dashed lines indicate the performance of BC policies trained on filtered sub-datasets consisting of top 5%, median 5%, and bottom 5% trajectories.
>
>
>
> ## Q: The paper notes that $\pi_{w}$ is initialized as $\pi_{D}$. Could you specify where this initialization is reflected in the algorithm box?
>
> A:   In the tabular setting, Proposition 1 establishes that the exact solution of Eq.9 remains within the data support when the actor is initialized as $\pi_\omega=\pi_{\mathcal{D}}$. However,  practical implementations often rely on a limited number of gradient descent steps for optimizing Eq.9. This leads to policy optimization errors which are further exacerbated iteratively. To overcome this issue, we find it is useful to also add the original behavior regularization. In summary, adding this term is used to help the policy remains within the data support, thus there is no further need to initialize  $\pi_\omega=\pi_{\mathcal{D}}$ in the practical algorithm.  In our experiments, we also didn't find clear differences of initializing or not initializing  $\pi_\omega=\pi_{\mathcal{D}}$.  Thus in the practical algorithm box, there is no need to perform the  initialization.

---

> > ### Author Response · Authors · 2023-11-22
> >
> > Dear reviewer, we have provided detailed responses, but have not yet hear back from you. We will appreciate it deeply if you could reply our rebuttal. We are sincerely looking forward to further discussions to address the reviewers concerns to our best. Thanks!

---

### Meta-Review · Area_Chair_KfUL · 2023-12-05

**Metareview:**

The submitte paper proposes a novel approach for offline reinforcement learning overcoming some issues of naive behavior regularization. In particular, the authors propose conservative policy iteration to iteratively refine the reference policy for behavior regularization. The algorithm is analyzed theoretical and evaluated empirically on the D4RL benchmark on which it achieves very promising results.

The paper addresses an important problem and proposes an easy-to-implement appraoch that shows significant performance improvements over several baselines in extensive experiments.
The reviwers raised concerns regarding clarity of the presentation, the experimental evaluation (hyperparameter tuning and universal benefits), and the soundness of the theoretical results. Unfortunately, not all reviewers took part in the discussion, so I am partly interpreting the authors response regarding these concerns. While additional experimental results likely have resolved the reviewers correspoding reviewers' concerns, concerns regarding clarity and theoretical soundness were not finally resolved/sufficiently addressed. Thus I am recommending the rejection of the paper. Nevertheless, I believe the paper can be turned into a strong paper by taking the reviewers' comments into account and updating the paper accordingly and I'd like to encourage the authors do so.

**Justification For Why Not Higher Score:**

Unresolved concerns which must be addressed before the paper should be accepted.

**Justification For Why Not Lower Score:**

n/a

---

### Decision · Program_Chairs · 2024-01-16

Reject